

# Machine learning with remote sensing data to locate uncontacted indigenous villages in Amazonia

Robert S. Walker[1] and Marcus J. Hamilton[2,3]

[1] Department of Anthropology, University of Missouri, Columbia, MO, USA
[2] Department of Anthropology, University of Texas at San Antonio, San Antonio, TX, USA
[3] Santa Fe Institute, Santa Fe, NM, USA

## ABSTRACT

**Background**. The world's last uncontacted indigenous societies in Amazonia have only intermittent and often hostile interactions with the outside world. Knowledge of their locations is essential for urgent protection efforts, but their extreme isolation, small populations, and semi-nomadic lifestyles make this a challenging task.

**Methods**. Remote sensing technology with Landsat satellite sensors is a non-invasive methodology to track isolated indigenous populations through time. However, the small-scale nature of the deforestation signature left by uncontacted populations clearing villages and gardens has similarities to those made by contacted indigenous villages. Both contacted and uncontacted indigenous populations often live in proximity to one another making it difficult to distinguish the two in satellite imagery. Here we use machine learning techniques applied to remote sensing data with a training dataset of 500 contacted and 25 uncontacted villages.

**Results**. Uncontacted villages generally have smaller cleared areas, reside at higher elevations, and are farther from populated places and satellite-detected lights at night. A random forest algorithm with an optimally-tuned detection cutoff has a leave-one-out cross-validated sensitivity and specificity of over 98%. A grid search around known uncontacted villages led us to identify three previously-unknown villages using predictions from the random forest model. Our efforts can improve policies toward isolated populations by providing better near real-time knowledge of their locations and movements in relation to encroaching loggers, settlers, and other external threats to their survival.

## INTRODUCTION

The ongoing colonization of Amazonia has brought waves of epidemics and violence for centuries with severe consequences for indigenous populations (*Bodard, 1974*; *Hemming, 1978*; *Hurtado et al., 2001*; *Hamilton, Walker & Kesler, 2014*). Amazingly, despite all the external pressures, remote areas in the upper Amazon watershed still support a number of remnant indigenous societies generally referred to as uncontacted or isolated populations (*Vaz, 2011*; *Castillo, 2004*; *Ricardo & Ricardo, 2011*). Despite these labels, intermittent and often hostile interactions with the outside world are commonplace (*Wallace, 2011*). Most

Corresponding author
Robert S. Walker,
walkerro@missouri.edu

governmental and non-governmental organizations promote no-contact policies for these isolated indigenous populations with the belief that they are safest if left to themselves (*Walker & Hill, 2015*). However, encroachment from loggers, miners, settlers, and others is incessant and uncontacted societies represent the world's most critically endangered cultures (*Walker, Kesler & Hill, 2016*). There is a need for good information on their locations and movements in hopes of improving their survival prospects moving forward.

Our project is part of a longitudinal remote surveillance program to conduct scientific studies of indigenous demography and spatial ecology to facilitate informed decisions by policy makers that will increase protection efforts for isolated indigenous populations (*Walker & Hamilton, 2014*; *Walker, Hamilton & Groth, 2014*). Our central goal is to gather as much information on isolated indigenous populations as possible without attempting any direct contact (*Kesler & Walker, 2015*). We maximize the use of available technologies to gather data remotely with no interference. Satellite imagery offers a safe, low-cost, and noninvasive method for studying population dynamics and spatial ecology of indigenous populations (*Walker, Kesler & Hill, 2016*). Similarly important is the need to understand spatial resource needs of indigenous societies in a region heavily impacted by deforestation, as well as the potential importance of connections among subpopulations, known to contribute to population viability (*Levins, 1969*; *Hanski, 1999*). The irreversible threats from large-scale habitat loss via deforestation and conversion of land to agriculture and pasture paint a bleak future for uncontacted populations (*Fagan & Shoobridge, 2005*; *Salisbury & Fagan, 2013*; *Walker, Kesler & Hill, 2016*). The hope is that better data and methods can contribute improvements to this complex issue.

Applied machine learning is a vital tool for conservation work as a means to both collect and analyze more data at faster rates (*Murray et al., 2018a*). The growing use of machine learning methods to analyze large sets of biological, biophysical, spectral and climatological data has enabled accurate differentiation of the world's landscapes (*Pettorelli et al., 2014*). More germane to our work are forest classification projects (*Hansen et al., 2013*; *Murray et al., 2018b*). The Global Forest Change dataset was developed by classifying pixels using 15 or more high-resolution global composite images as predictors, each developed from over 500,000 Landsat images (*Hansen et al., 2013*).

The random forest algorithm is known to give excellent classification results and relatively quick processing speed (*Du et al., 2015*; *Pal, 2005*; *Rodriguez-Galiano et al., 2012*). Random forests (*Breiman, 2001*) are an ensemble supervised learning method that builds multiple decisions trees used here for the classification of village class (uncontacted versus contacted). Random forests operate by constructing a multitude of decision trees. Some of the advantages of random forests are that they are robust to inclusion of features that are irrelevant to classification, and they are invariant to transformations of feature variables (*Belgiu & Drăguț, 2016*). For these reasons, the random forest algorithm is popular for remote sensing data given its accuracy, speed, and ability to handle high data dimensionality and multicollinearity.

## MATERIALS & METHODS

### Data

We combined the exact locations (centroids) of 25 uncontacted and 500 contacted indigenous villages (*Walker, Kesler & Hill, 2016*). More information about our general project along with high-resolution imagery for uncontacted villages is available at https://isolatedtribes.missouri.edu. The locations of uncontacted villages were originally derived from scouring high-resolution imagery using a combination of undergraduate helpers and various maps made by governmental and non-governmental agencies in Colombia, Ecuador, Peru and especially Brazil. Several additional locations have been pieced together from governmental reports and news stories stemming from overflights. Contacted villages are from the Brazilian government website (http://www.funai.gov.br/), and we included all of those that were in western Amazonia (west of 60 degrees longitude, Fig. 1).

Hansen and colleagues' (*2013*) Global Forest Change (GFC) project provides small-scale deforestation at approximately 30 m resolution from Landsat sensors extending back to the year 2000. GFC version 1.5 goes up through the year 2017. We extracted the amount of detected deforestation in $2 \times 2$ km squares surrounding each village's centroid and took the maximum area cleared in any one particular year from across the 17-year period. We refer to this measure as cleared area as it includes both the village and associated gardens but not those of neighboring villages. In addition, our dataset has other features, including regional population density in the nearest 100 square km (*CIESIN, 2005*), elevation at 30 m digital resolution from the Space Shuttle Radar Topography Mission (*Rabus et al., 2003*), and distance to populated places at 10 m resolution (*Balk et al., 2006*). We also included a local lights-at-night measure at 3 km resolution (*Pritchard, 2017*, from https://earthobservatory.nasa.gov) using the distance from village centroid to the nearest detected lights. Finally, distance to rivers of all the different Strahler stream orders using the Global Self-consistent, Hierarchical, High-resolution Geography Database (*Wessel & Smith, 1996*), along with the minimum distance to combined rivers of Strahler stream orders 1, 2, and 3, giving a total of 11 features used to train algorithms.

### Models

Machine learning algorithms were performed with the R package caret. We found that an untuned random forest algorithm had a fairly high combination of sensitivity (true positive rate) and specificity (true negative rate) in the 0.8 to 0.9 range. As mentioned above, random forest algorithm is an ensemble classifier that produces multiple decision trees, using a randomly selected subset of training samples and variables. Other algorithms such as neural networks, extreme gradient boosting tree, and lasso logistic regression were also relatively-high performing but gave slightly lower values on one or the other metric.

The target classes in our sample are imbalanced with only 4.8% of villages in the sample being uncontacted. During model training we noticed that varying the detection cutoff (also known as the threshold) that classifies villages into one class or the other had large effects on the results (the default cutoff is 0.5 majority rule). In addition, common loss

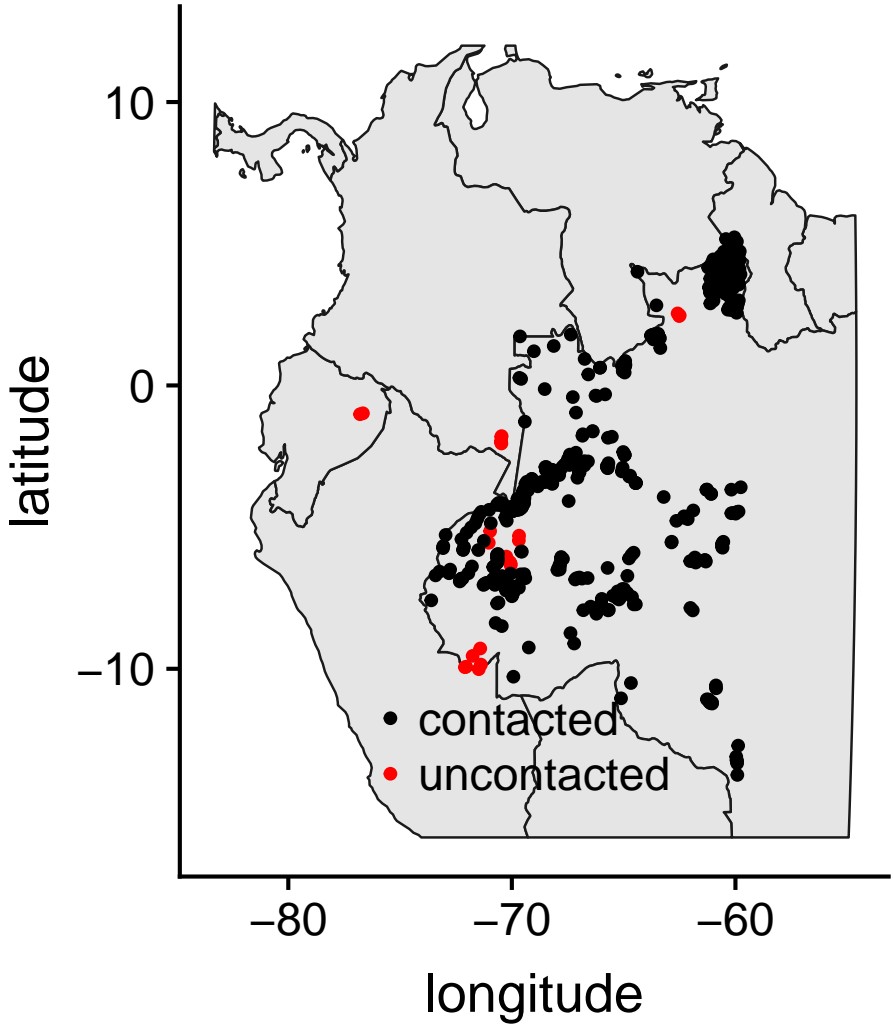

**Figure 1 Map of study locations.** Map of 500 contacted indigenous villages in Brazil and 25 uncontacted indigenous villages in Brazil, Colombia, Ecuador, and Peru that were included in the study.

metrics such as the area under the ROC curve or the F1 score tended to give either high specificity or sensitivity with our data, but not both.

To address the imbalanced data issue and improve model performance, we used a random forest algorithm that iteratively tuned the cutoff value such as to simultaneously maximize both specificity (true negative rate) and sensitivity (true positive rate). In other words, we instituted cost-sensitive learning into the random forest (*Elkan, 2001*; *Zadrozny, Langford & Abe, 2003*; *Khoshgoftaar, Golawala & Hulse, 2007*). The loss metric we used for training is the distance from a perfect model of sensitivity of 1 and specificity of 1. We used 1,000 trees with 2 variables available for splitting at each tree node. To evaluate models we used a leave-one-out cross-validation (non-nested) looped over a range of cutoffs from 0.01 to 0.99 in increments of 0.01. Raising the cutoff value means a higher level of evidence (i.e., more decision trees out of the total 1,000 trees that comprise the random forest) is

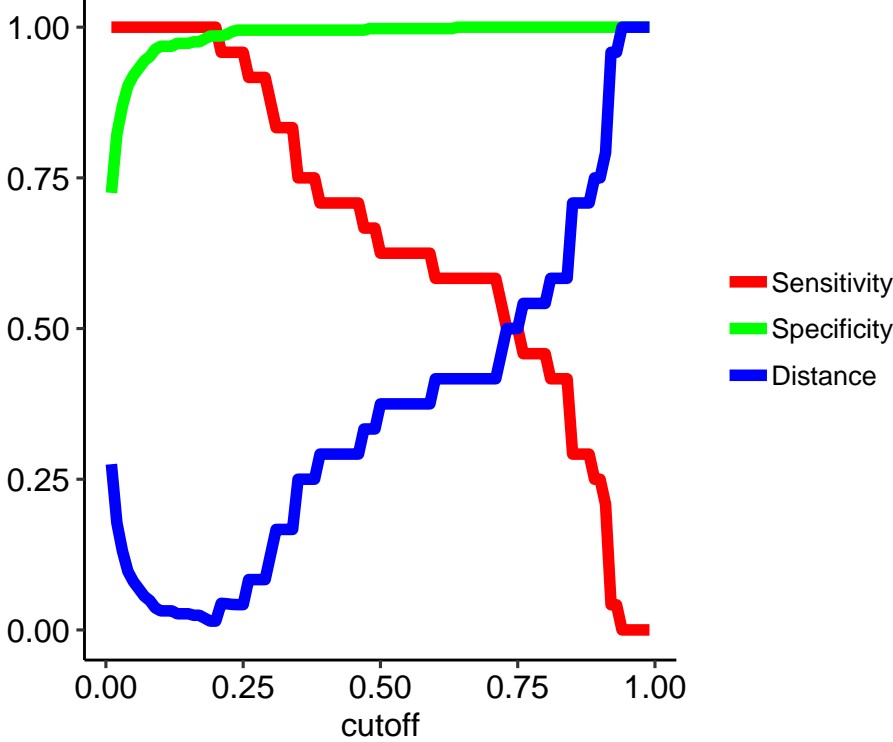

**Figure 2** **Model metrics obtained from training the random forest model across a range of cutoffs from 0.01 to 0.99 in increments of 0.01.** To train the random forest model we used leave-one-out cross-validation across a range of cutoffs from 0.01 to 0.99 in increments of 0.01. Raising the cutoff value means a higher level of evidence is needed to assign the positive class (uncontacted), which decreases sensitivity (true positive rate) and increases specificity (true negative rate). Here the optimal cutoff (0.2) gives a perfect cross-validated sensitivity of 1.0 and a specificity of 0.98. The distance is the distance from a perfect model which is minimized during training.

needed to assign the positive class (uncontacted) so it decreases sensitivity and increases specificity. Here a sensitive cutoff of 0.2 yields a minimal distance metric and the desired combination of high sensitivity and specificity metrics (Fig. 2).

## RESULTS

Our random forest algorithm, with an optimally-tuned cutoff of 0.2, yields a sensitivity of 1.0 and a specificity of 0.98 using leave-one-out cross-validation. This means that all uncontacted villages are correctly classified and 98% of the contacted villages are correctly classified. Therefore, our model has a strong ability to automatically distinguish between contacted and uncontacted villages. In order of descending variable importance, uncontacted villages have (1) smaller cleared areas, (2) longer distances from lights, (3) higher elevation, (4) longer distances to populated places, (5) lower regional population density, (6) longer distances from rivers of all Strahler stream orders up to and including 3, and (7) shorter distances to rivers of levels 4 and 5. Figure 3 shows density plot comparisons for the top 4 features in terms of variable importance.
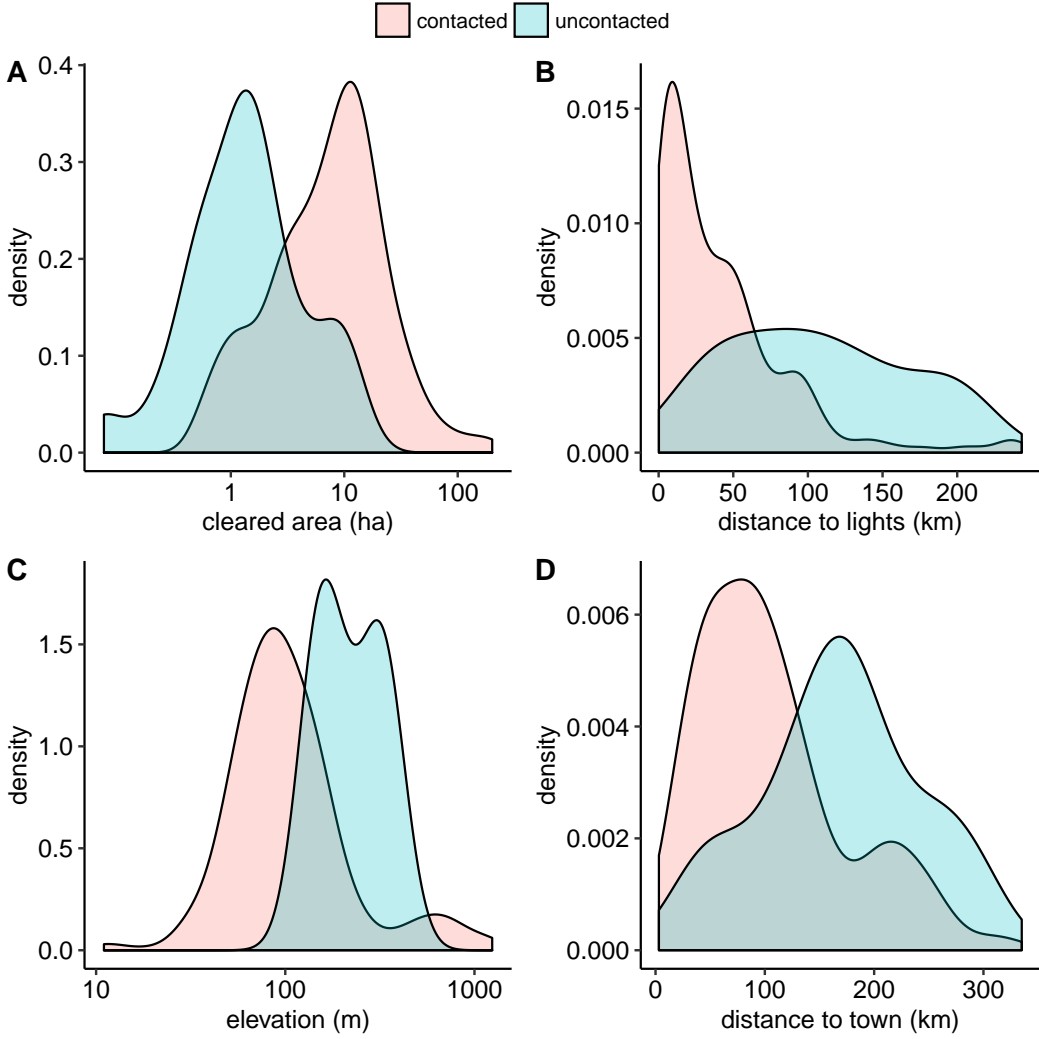

**Figure 3** **Smoothed kernel density plots comparing uncontacted to contacted indigenous villages.** The top four distinguishing features in terms of variable importance in the random forest model are uncontacted villages have (A) smaller cleared areas, (B) farther distances to satellite-detected lights at night, (C) higher elevation, and (D) farther distances to populated places, on average. A and C are best visualized on log scales.

Given the success of our algorithm during cross-validation, we then moved to implement it for predictive purposes. We did a grid search of all 2 × 2 km squares within a 100 km radius of the five clusters of known uncontacted villages (Fig. 1). This approach does produce a high number of false positives created by natural clearings (e.g., landslides, windfalls, etc.). Fortunately, most natural clearings can be eliminated by simply removing all clearings that are less than 0.5 ha. This left us with a sample of 20 clearings. Of these we were able to obtain high resolution imagery for eight of these and three contained newly-identified villages. One of these in Colombia appears to be currently inhabited given that it has a single longhouse structure and shows recently made clearings in Global Land

Analysis and Discovery (GLAD, *Tyukavina et al., 2016*). The GLAD alert system processes Landsat imagery as it becomes available to identify tree cover change in near real-time. This is an invaluable system for monitoring both recent activity by uncontacted villages, as well as encroaching deforestation from outsiders.

The other two newly-discovered sites are historical villages. One is from the uncontacted Yanomami in northern Brazil inhabited from around year 2000 or earlier and until 2004. The other is from Pano speakers on the border between Peru and Brazil and was probably inhabited during a similar time period. The other five possible locations identified by the random forest predictions with high resolution imagery available all appeared to be natural. Therefore, we estimate our testing precision with this small sample as 0.375 (3 true positives divided by 8 total cases).

## DISCUSSION

We used deforestation data from Landsat satellites to train algorithms to identify the locations of uncontacted indigenous groups in Amazonia as part of an ongoing effort to better understand their conservation status and threats. Our results show that uncontacted villages have smaller cleared areas, reside at higher elevations, and are farther from populated places and satellite-detected lights at night. Our random forest algorithm with an optimally-tuned cutoff has cross-validated performance metrics of over 98%.

The case of the uncontacted Yanomami (also known as the Moxihatetea) is a good example of the importance of a near real-time monitoring system. Their previous village was abandoned in late 2014 and the Brazilian indigenous agency (FUNAI) and the Yanomami indigenous association (Hutukara) were particularly worried that some disaster had befallen them since much of the nearby area has seen invasions by gold miners. For a year and a half their whereabouts were unknown. We began looking for them using Landsat data, but it was the remote sensing fire alerts (FIRMS, *Davies et al., 2009*) that first alerted us to their exact location. We tasked a DigitalGlobe satellite image on May 12, 2016 and were relieved to find out that they were alive and well and clearing large gardens. The number of sections in their *shabono* village structure had increased from 16 to 17. We relayed this information on to FUNAI and Hutukara who then organized a flyover to officially confirm the location.

Remote sensing provides many advantages over flyovers, and we actually do not recommend them. As we have shown, the information provided solely by remote sensing is sufficient to identify uncontacted villages. Remote sensing is safe, low-cost, and noninvasive, while flyovers are not. Population estimates are also crucial information for assessing trends in the demographic health of isolated populations by measuring areas of fields, villages, and houses in satellite imagery. Heads-up digitization of satellite imagery provides better population estimates than do flyovers where most people are not visible because many hide or run away in fear. Remote sensing offers the benefits of time-stamped evidence of occupation of areas inhabited by isolated populations, along with movements through time (*Walker, Kesler & Hill, 2016*).

## CONCLUSIONS

A dozen easily obtainable remote sensing measures allowed our random forest algorithm to successfully classify uncontacted versus contacted villages. Extending the algorithm to make predictions in a grid search greatly accelerates our ability to find and identify the locations of uncontacted villages. Moving forward we anticipate using an even lower cutoff value because the decreasing costs in satellite imagery make false positives from a more sensitive algorithm relatively cheap to evaluate and discard. We anticipate that this method will become the primary means by which to track and locate these same uncontacted villages, as well as undiscovered locations of uncontacted villages.

One shortcoming of our classification model when applied to searching through unlabeled satellite imagery is that it was not designed to classify natural landslides, windfalls, or riverbank clearings. All of these natural processes also create deforestation signatures that further complicate our searches. Future work could well include these, but in the meantime we filter our predictions based on cleared area because natural clearings tend to be less than 0.5 ha while most uncontacted villages have larger areas than that.

Our research is vital and timely as isolated groups are among the last remaining small-scale subsistence populations living in a traditional lifestyle. The enormous and mounting pressure from external threats create the possibility that isolated populations will disappear in the near future. Better monitoring and tracking with remote sensing are tools that might provide more informed conservation decisions concerning increased protection and land rights for the world's most critically-endangered human cultures.

## ACKNOWLEDGEMENTS

We thank Mark Flinn and the Comparative Methods course at the University of Missouri for their help and suggestions.

### Funding

This work was supported by a National Geographic Society Research and Exploration Grant (#9764-15). The funders had no role in study design, data collection and analysis, decision to publish, or preparation of the manuscript.

### Grant Disclosures

The following grant information was disclosed by the authors:
National Geographic Society Research and Exploration Grant: #9764-15.

### Competing Interests

The authors declare there are no competing interests.

### Author Contributions

- Robert S. Walker and Marcus J. Hamilton conceived and designed the experiments, performed the experiments, analyzed the data, contributed reagents/materials/analysis

tools, prepared figures and/or tables, performed the computation work, authored or reviewed drafts of the paper, approved the final draft.

## Data Availability

The raw remote sensing variables are available in the Supplemental File.

## Supplemental Information

Supplemental information for this article can be found online at http://dx.doi.org/10.7717/peerj-cs.170#supplemental-information.

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
