# Peer review of "Machine learning with remote sensing data to locate uncontacted indigenous villages in Amazonia"

_PeerJ Computer Science, doi:10.7717/peerj-cs.170_

## Round 0.1 · original submission · Minor Revisions

Dear authors,

Many thanks for your contribution.

The reviewers recognize that your work is relevant, well-written and well-organized.

However, they raised a number of points that need to be better explained and clarified.

When preparing your revised manuscript, you are asked to carefully consider the reviewers’ comments, and submit a list of responses to the comments.

Reviewer 1 ·

Basic reporting

no comment

Experimental design

no comment

Validity of the findings

no comment

Additional comments

This is an excellent piece of work aiming at locating uncontacted indigenous population in Amazonia with non-invasive technology, namely random forest applied on Landsat imagery. I would recommend accepting the manuscript as it is. I identified only an error: line 109- Wessel & Smith, 1986 (1996 in the reference list). While "Strahler numbers" is an accepted name, I would suggest replacing with "Strahler stream order" throughout the manuscript, as the later is less ambiguous.

Reviewer 2 ·

Basic reporting

No comment.

Experimental design

No comment.

Validity of the findings

There is only a point where I would appreciate a clearer statement.

Experiments are based on the leave-one-out (LOO) paradigm due to the limited data set size. Moreover, it clearly emerges from the analysis that the results in the classification between contacted and uncontacted villages strongly depends on the random forest parameters, in particular on cutoff. Therefore, the procedure applied to estimate this parameter is crucial to evaluate performance: from the text, it is not clear if two nested LOO paradigms have been used to optimize its value and evaluate performance. If this is the case, please state it clearly. On the other hand, if a LOO experiments has been repeated by using different threshold values, then performance would be overestimated and there is a risk of overfitting, also because the data set is so small. A similar clarification is also needed for the task of predicting new villages.

Reviewer 3 ·

Basic reporting

I really enjoyed reading this paper. It is well-written, well-organized and provides sufficient background information.

Experimental design

My only criticism is related to the explanation of the created prediction model. There is no information on how the training samples were collected. I assume that this information is available on the project website. However, I think that the readers have to be informed in the paper itself about this. Furthermore, the variables used as input for creating the model are not described. For example, where did you get the data on the lights at nigh fromt? Is this variable related to the distance to town variable? Why did you decide to use these eleven specific variables to create the prediction model? How many decision trees did you defined, i.e. ntree parameter? How many variables did you use to split the trees nodes, i.e. mtry parameter?
Do the uncontacted villages share the same characteristics? I am not an expert in this field, but I am curious whether there are specific cultural traits that might influence the relevance of the variables used to predict the presence of these villages. For example, the four variables identified as being the most relevant variables might vary from one village category to another??
Specific comments are available below:
Line 35: How do you define near-real time in your study?
Line 83: how did you collect the data on the uncontacted and contacted villages?
Lines 106-107: how did you identify the lights-at-night?
Lines 113: which preliminary work do you refer to?


Figures: I am not sure why there is a duplication of the figures’ explanations? (figures captions)
Figure 2: please check the leave-one-out (you wrote leave-outout)_

Validity of the findings

The reported results have a clear and important societal relevance. The impact of the study is well described.

Additional comments

no further comments

---

## Round 0.2 · accepted · Accept

Dear authors,

The reviewers ’comments have been addressed.

I think that your work is well-suited for PeerJ Computer Science readers.

Many thanks for your contribution.

Reviewer 2 ·

Basic reporting

No comment.

Experimental design

No comment.

Validity of the findings

No comment.

Additional comments

I think the paper is now ready for publication.